# BRCA1 deficiency specific base substitution mutagenesis is dependent on translesion synthesis and regulated by 53BP1

Dan Chen[1,9], Judit Z. Gervai[1,9], Ádám Póti[1], Eszter Németh[1], Zoltán Szeltner[1], Bernadett Szikriszt[1], Zsolt Gyüre[1,2], Judit Zámborszky[1], Marta Ceccon[3], Fabrizio d'Adda di Fagagna [3,4], Zoltan Szallasi [5,6,7], Andrea L. Richardson [8✉] & Dávid Szüts [1✉]

Defects in *BRCA1*, *BRCA2* and other genes of the homology-dependent DNA repair (HR) pathway cause an elevated rate of mutagenesis, eliciting specific mutation patterns including COSMIC signature SBS3. Using genome sequencing of knock-out cell lines we show that Y family translesion synthesis (TLS) polymerases contribute to the spontaneous generation of base substitution and short insertion/deletion mutations in *BRCA1* deficient cells, and that TLS on DNA adducts is increased in *BRCA1* and *BRCA2* mutants. The inactivation of *53BP1* in *BRCA1* mutant cells markedly reduces TLS-specific mutagenesis, and rescues the deficiency of template switch–mediated gene conversions in the immunoglobulin V locus of *BRCA1* mutant chicken DT40 cells. 53BP1 also promotes TLS in human cellular extracts in vitro. Our results show that HR deficiency–specific mutagenesis is largely caused by TLS, and suggest a function for 53BP1 in regulating the choice between TLS and error-free template switching in replicative DNA damage bypass.

[1] Institute of Enzymology, Research Centre for Natural Sciences, Budapest H-1117, Hungary. [2] Doctoral School of Molecular Medicine, Semmelweis University, Budapest H-1085, Hungary. [3] IFOM Foundation-FIRC Institute of Molecular Oncology Foundation, Via Adamello 16, 20139 Milan, Italy. [4] Istituto di Genetica Molecolare, Consiglio Nazionale delle Ricerche (IGM-CNR), Via Abbiategrasso 207, 27100 Pavia, Italy. [5] Computational Health Informatics Program (CHIP), Boston Children's Hospital and Harvard Medical School, Boston, MA 02215, USA. [6] Danish Cancer Society Research Center, Copenhagen 2100, Denmark. [7] SE-NAP, Brain Metastasis Research Group, 2nd Department of Pathology, Semmelweis University, Budapest H-1092, Hungary. [8] Johns Hopkins University School of Medicine, Baltimore, MD 21287, USA. [9] These authors contributed equally: Dan Chen, Judit Z. Gervai. ✉email: aricha58@jhu.edu; szuts.david@ttk.hu

Defects in the *BRCA1* tumour suppressor gene are associated with genomic instability. Inherited mutations of *BRCA1* increase the risk of breast and ovarian cancer[1], and somatic biallelic inactivating *BRCA1* mutations are also commonly seen in cancer cells[2]. The analysis of tumour whole-exome or whole-genome sequences revealed associations between specific mutation patterns and the loss of *BRCA1* function. These comprise a broad-spectrum base substitution signature termed SBS3, short insertions and deletions (indels), and certain types of large rearrangements[3–6]. Experiments with isogenic knock-out cell lines proved the causative role of *BRCA1* loss in these mutagenic processes, and also allowed the measurement of increased mutation rates in *BRCA1* deficient cells[7]. Mutational analyses of tumours and knock-out cell lines indicated that the inactivation of *BRCA2*, *PALB2* and other genes encoding proteins responsible for homology-dependent repair (HR) also elicits SBS3 signature mutations, but induces different patterns of indels[2,7,8]. Although the genomic scars associated with HR deficiency (HRD) are useful biomarkers for cancer treatment[9,10], there is a distinct lack of understanding of the molecular mechanism behind the formation of SBS3 mutations.

BRCA1 is a large, multifunctional protein that forms a stable complex with BARD1[11]. BRCA1 mediates the recombination-based repair of DNA double-strand breaks (DSBs) in several ways. It promotes the nucleolytic resection of the 5′ broken strand through binding to CtIP[12,13], and subsequently also promotes the recruitment of RAD51 to the 3′ single-stranded DNA tails in association with PALB2 and BRCA2[14,15], and by directly binding RAD51 and enhancing its recombinase activity[16]. BRCA1 also participates in competitive regulation of DSB repair in antagonism with 53BP1[17,18] and associated protein complexes, which inhibit end resection and promote canonical non-homologous end joining (NHEJ)[19]. Linked to its ability to ubiquitylate histone H2A, BRCA1 can prevent the recruitment of 53BP1 to DSBs[20]. The loss of DSB end protection due to the disruption of *53BP1* reactivates HR in *BRCA1* mutant cells and attenuates their sensitivity to inhibitors of topoisomerase I or poly-ADP ribose polymerase 1 (PARP-1)[18,21]. BRCA1 and BRCA2 also help protect stalled and reversed replication forks through recruiting RAD51 to the exposed nascent strands, and thereby antagonising their degradation promoted by the MRN complex and PTIP, but not 53BP1[22–24].

An increased use of NHEJ in *BRCA1* mutants offers a straightforward explanation for the arising genomic deletions and rearrangements, but not the single base substitutions (SBSs). An important general cause of base substitution mutagenesis is the replication of damaged DNA by the process of translesion DNA synthesis (TLS). Specialised TLS polymerases are able to utilise damaged DNA templates, but are prone to introducing mutations due to their low fidelity or the misinstructional nature of the lesions[25,26]. There also exists an error-free mode of damage bypass that uses the nascent strand of the sister chromatid as a lesion-free alternative template. This recombination-mediated template switching mechanism relies on the recombinase Rad51 and its paralogues or Rad52 in *Saccharomyces cerevisiae*[27,28], and the use of the sister chromatid as template was found to depend on BRCA1 and the RAD51 paralogue XRCC3 in vertebrate cells[29]. BRCA1 and other HR factors therefore appear to have a central role in the error-free replicative bypass of DNA lesions.

We thus hypothesised that the majority of mutagenesis in *BRCA1* deficient cells, especially base substitution mutagenesis, is the consequence of an increased use of TLS that results from the loss of the template-switching bypass pathway. We show that the inactivation of translesion DNA polymerases reduces base substitution and small indel mutagenesis in the absence of BRCA1, and that the bypass of DNA adducts is more mutagenic in *BRCA1*

mutant cells. Importantly, the deletion of *53BP1* reduces TLS-dependent mutagenesis and appears to restore template switching in *BRCA1* deficient cells, which points to a central role of 53BP1 in controlling pathway choice in replicative DNA damage bypass.

## Results

**TLS makes an important contribution to mutagenesis associated with BRCA1 deficiency.** TLS primarily uses polymerases of the Y family, plus the B family polymerase (Pol) ζ[25]. To test whether TLS is responsible for mutagenesis associated with HRD, we created double mutant DT40 cell lines by homologous gene targeting, disrupting *BRCA1* in *POLH*$^{-/-}$, *POLK*$^{-/-}$ and *REV1*$^{-/-}$ cell lines that carry homozygous deletions in the Y family polymerases Polη, Polκ and REV1, respectively. As the avian genomes do not code for an orthologue of human Polι, our cell line panel included all possible Y family polymerases. We measured spontaneous mutagenesis in these cell lines as well as the matching wild-type (WT) and single mutant controls by culturing a single cell-derived ancestral clone until a second cloning step 50 days (approx. 100 generations) later, and comparing the whole-genome sequences of the ancestral clone and at least three descendent clones. Mutations were identified using the IsoMut tool (Supplementary Data 1–4), which was developed for high-efficiency detection of base substitutions and short insertions and deletions (indels) in isogenic samples[30]. Confirming our earlier results[7], we found a significantly higher number of SBSs, short insertions and short deletions in *BRCA1*$^{-/-}$ cells compared to WT cells (Fig. 1a–c). SBS mutagenesis was lower in *BRCA1*$^{-/-}$ *POLH*$^{-/-}$ cells ($p = 0.004$, t-test), *BRCA1*$^{-/-}$ *POLK*$^{-/-}$ cells ($p = 0.055$), and *BRCA1*$^{-/-}$ *REV1*$^{-/-}$ cells ($p = 0.0001$) than in *BRCA1*$^{-/-}$ controls, suggesting that Y family TLS polymerases are responsible for at least part of the increased base substitution mutagenesis in *BRCA1* deficient cells. A similar reduction of the number of insertions and deletions in the double mutants also suggests TLS involvement in the generation of indels, though the number of events was small and not all changes were significant (Fig. 1b, c). The involvement of TLS polymerases in HRD-associated mutagenesis does not appear to be caused by their increased expression, as we found no difference in their transcript levels between WT and *BRCA1*$^{-/-}$ cells (Supplementary Fig. 1).

SBS mutations can be presented as triplet spectra that take identity of the preceding and the following nucleotide into account (Supplementary Fig. 2). To gain a better understanding of the spontaneous SBS spectra, we attempted to decompose them into a small number of mutational signatures using non-negative matrix factorisation (NMF). We obtained good decomposition with three signatures (Supplementary Fig. 3, Supplementary Data 5). Signature 1A dominated the WT and the single polymerase mutant spectra, and was similar to the background (BG) signature we described for the DT40 cell line earlier (Fig. 1d, e, Supplementary Fig. 4a)[8]. Signature 1B was similar to the HRD signature we described[8] and to COSMIC signature SBS3 (Supplementary Fig. 4b, Supplementary Data 6), which is associated with *BRCA1/2* mutant cancers[6]. Signature 1B explained the majority of the extra mutations in *BRCA1* mutant cell lines compared to their controls, except for the *BRCA1*$^{-/-}$ *POLK*$^{-/-}$ cell line, in which an additional Signature 1C was needed to reconstruct base substitutions. Signature 1C is dominated by T > A mutations in a (C/G/T)TT context, and it appears to correlate with the inactivation of the *POLK* gene, as it also appears in the single *POLK*$^{-/-}$ mutant (Fig. 1d, e). In a deconstruction of whole-genome mutation datasets from the pan-cancer PCAWG database into all COSMIC v3.1 SBS signatures plus Signature 1C (now named Sig.POLKΔ, for POLK deficiency), the experimentally defined signature appeared as a minor component of a number of samples derived primarily from

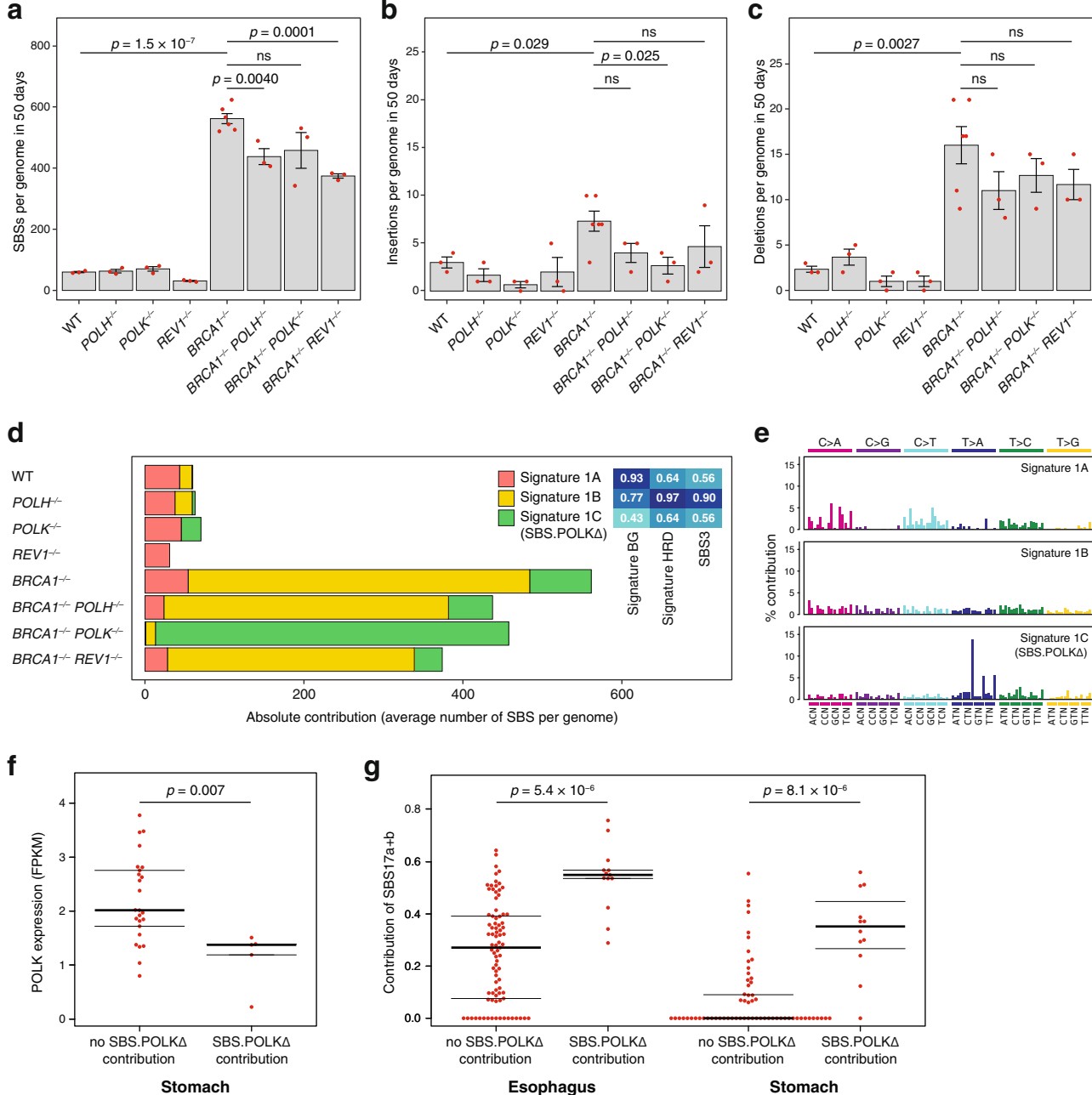

**Fig. 1 TLS plays a role in spontaneous mutagenesis in BRCA1 deficient cells.** The mean number of newly arising SBS (single base substitution, **a**), insertion (**b**) and deletion (**c**) mutations detected in DT40 cell clones of the indicated genotypes following 50 days of culturing an ancestral clone. Error bars indicate standard error of the mean (SEM), individual values are also shown with red symbols. The significance of tested differences is shown (ns not significant, unpaired two-sided $t$-test, $n = 3$ biologically independent samples except n = 6 for $BRCA1^{-/-}$ cells). **d** Deconstruction of the averaged spontaneous triplet SBS spectra of the indicated genotypes into three mutational signatures using non-negative matrix factorisation. The inset shows cosine similarities to published triplet signatures, see text for details. **e** Triplet spectra of the mutational signatures derived in (**d**). Each mutation class, as indicated at the top of the panel, is separated into 16 categories based on the identity of the preceding and following nucleotide as shown below. The order of the following nucleotides, not shown due to lack of space, is alphabetical. **f** POLK expression level in all PCAWG stomach adenocarcinoma samples with RNAseq data, grouped according to whether >5% contribution of SBS.POLKΔ was detected in a deconstruction of the somatic mutation spectrum to all COSMIC v3 SBS signatures. **g** The summed contribution of COSMIC triplet signatures SBS17a and SBS17b to the somatic mutation spectrum in all PCAWG oesophagus or stomach adenocarcinoma samples, grouped according to SBS.POLKΔ contribution as in (**f**). The mean, upper and lower quartiles are shown in (**f**) and (**g**), statistical significances are indicated (Wilcoxon rank-sum test). Source data are provided as a Source Data file.

oesophageal and gastric cancer (Supplementary Figs. 5 and 6). RNA sequencing–derived gene expression data was available for many of the gastric cancer samples, and showed significantly lower POLK expression in samples with a Sig.POLKΔ component in their SBS spectrum (Fig. 1f). The presence of Sig.POLKΔ

strongly correlated with the proportion of mutations that belong to SBS17a + SBS17b (Fig. 1g). SBS17 has been hypothesised to arise due to oxidative damage[31], and it is plausible that the absence of Polκ alters the spectrum of the resulting mutations. In any case, the finding that the inactivation of a TLS gene can

change the spectrum of mutagenesis in *BRCA1* deficient cells further supports the role of the TLS pathway in HRD-specific mutagenesis.

**Replication-blocking lesions are more commonly bypassed by TLS in HR deficient cells.** In the case of spontaneous mutagenesis, the identity of the potential mutagenic DNA lesions is not known. We attempted to learn more about the causes of mutagenesis in HR deficient cells using treatments with cisplatin, a well-known mutagen that primarily causes intrastrand adducts at GG or AG dinucleotides[32] which block the progress of replicative polymerases[33]. The majority of cisplatin-induced mutations occur at these sequence motifs[34–36], and are thought to be the consequence of translesion DNA synthesis[37,38], which contributes to cisplatin resistance in cancers[39]. We exposed WT, *BRCA1*$^{-/-}$ and *BRCA2*$^{-/-}$ DT40 cells to four weekly rounds of 1 h cisplatin treatments. Compared to the mock treatment of equivalent total duration (50 days), cisplatin-induced extra base substitutions in all three cell lines, but the increment was greater in the mutants than in the WT (Fig. 2a). An NMF decomposition of the SBS spectra once again yielded signatures that describe spontaneous BG and HRD-specific mutagenesis (Signatures 2A and 2B), and a third signature responsible for the majority of mutations in cisplatin-treated samples (Signature 2C, Fig. 2b, c, Supplementary Fig. 4, Supplementary Data 5). Importantly, the entire increment compared to the mock treatments was explained by mutations that belong to the cisplatin-specific signature, and cisplatin did not cause HRD signature mutations in the HR deficient cells. This led us to two conclusions: (1) more TLS takes place in *BRCA1/2* mutants upon treatment with the same dose of cisplatin, also supported by the higher GG-specific NCC > NAC peaks in the triplet spectra (Fig. 2d), (2) mutations that belong to the spontaneous HRD spectrum are not caused by mechanisms that are also expected to be triggered by cisplatin treatment, for example DNA synthesis at reversed replication forks.

Polκ is responsible for the majority of cisplatin-induced base substitutions in human cells[40], so to support our conclusions on increased TLS in *BRCA1*$^{-/-}$ cells, we also subjected *BRCA1*$^{-/-}$ *POLK*$^{-/-}$ cells to equivalent cisplatin treatments. The mean number of total SBS mutations was lower in the double mutants than in *BRCA1*$^{-/-}$ cells (985 vs. 1514, Supplementary Data 4). More importantly, the number of cisplatin-specific NCC > NAC mutations was almost as low in the *BRCA1*$^{-/-}$ *POLK*$^{-/-}$ cells as in the WT control (Fig. 2e), showing that the incremental mutagenicity of cisplatin in BRCA1 deficient cells is due to the TLS enzyme Polκ.

Cisplatin was also more mutagenic in *BRCA1*$^{-/-}$ and *BRCA2*$^{-/-}$ cells than in the WT with respect to small insertions and deletions (Fig. 2f, i). In agreement with earlier reports, we found that the most common cisplatin-induced insertions were GGT > GGTT single T insertions (Fig. 2g), which are expected to result from TLS on GG intrastrand crosslinks[34,35]. These insertions, which seldom arise in untreated cells, were five-fold more numerous in cisplatin-treated *BRCA1*$^{-/-}$ cells than in WT cells, though interestingly they were at the WT level in *BRCA2*$^{-/-}$ cells (Fig. 2h). The extra cisplatin-induced deletions in *BRCA1/2* mutants were almost all 1 or 2 bp long (compare the panels in Fig. 2j). Such very short deletions mostly occur at potential sites of cisplatin adducts[34], thus these are also likely to arise from error-prone TLS. Taken together, the higher number of short insertions and deletions at potential sites of cisplatin lesions supports the conclusion of an increased use of TLS in *BRCA1/2* mutants.

**53BP1 influences mutagenic damage bypass in BRCA1 deficient cells.** BRCA1 and other HR proteins possess a function in the repair of DSBs, offering an error-free alternative to NHEJ. We created double mutant *BRCA1*$^{-/-}$ *KU70*$^{-/-}$ and *BRCA1*$^{-/-}$ *53BP1*$^{-/-}$ DT40 cell lines to assess the role of NHEJ in mutagenesis in *BRCA1* deficient cells. The measurement of spontaneous mutagenesis by whole-genome sequencing showed a very substantial reduction of mutation numbers upon inactivating *53BP1* in a *BRCA1*$^{-/-}$ BG, with 71% fewer base substitutions, 59% fewer insertions and 69% fewer deletions (Fig. 3a–c) in the double mutant. The deletion of the gene for the NHEJ core factor KU70 caused a moderate decrease of base substitutions, and no significant decrease of insertions in the *BRCA1*$^{-/-}$ BG, suggesting that the anti-mutagenic effect of *53BP1* deletion goes beyond the role of this protein in NHEJ. The deletion of either *53BP1* or *KU70* similarly decreased the number of deletions, with an especially strong effect on deletions of three base pairs or longer, which presumably arise from NHEJ activity (Fig. 3c).

We also measured mutagenesis in the HR and/or NHEJ deficient cell line panel upon cisplatin treatment in order to directly assess any effect of the gene mutations on TLS. The inactivation of *53BP1* significantly lowered the number of events in all mutation classes in the *BRCA1*$^{-/-}$ background, although it had limited effect in the WT background (Fig. 3d–f). In contrast, there was no significant difference in the number of cisplatin-induced base substitutions, insertions or deletions between the *BRCA1*$^{-/-}$ and *BRCA1*$^{-/-}$ *KU70*$^{-/-}$ cell lines. In summary, our results point to an unexpected, specific role of 53BP1 in promoting TLS in *BRCA1* mutant cells.

**53BP1 antagonises HR-mediated template switching bypass.** In replicative DNA damage bypass, the alternative to TLS is homology-dependent template switching, which is difficult to detect due to its non-mutagenic nature. For the simultaneous detection of TLS and template switching we turned to a natural genomic 'lesion bypass reporter' of DT40 cells, the constitutively diversifying immunoglobulin light chain variable gene. In this locus, activation-induced deaminase (AID) converts cytosines to uracils, which are then excised to produce replication-stalling abasic sites. These sites can be bypassed by TLS to result in somatic hypermutation at C/G base pairs, or they can induce gene conversion (GC) templated by a set of upstream homeologous pseudogenes, which is a special example of HR-mediated template switch that results in detectable templated mutations[41]. We developed a variant of this assay, whereby we isolated multiple single-cell clones from WT and mutant cell lines, and after 40 days of culturing we analysed the arising new mutations in each total unsorted population using high coverage amplicon sequencing of the centre 233 bp of the variable VL1 region (Fig. 4a). The advantage of this approach is the ability to detect all mutations irrespective of their effect on the encoded protein, even at very low frequencies. However, it is a disadvantage compared to earlier Sanger sequencing methods that we were unable to entirely eliminate base substitution errors resulting from PCR and sequencing. As a result, we detected point mutations at A/T base pairs in a genotype-independent manner, which, together with an unknown number of mutations at C/G, must be false positives. Nevertheless, the genotype-dependent differences in mutation numbers were very informative. Confirming earlier results[42,43], there were significantly more point mutations at C/G base pairs and fewer GCs in the *BRCA1*$^{-/-}$ cell line than in the WT, whereas there were fewer C/G point mutations and more GCs in the *REV1*$^{-/-}$ cell line (Fig. 4b). The disruption of *53BP1* did not significantly affect base substitution mutagenesis, but it significantly increased the number of GCs in the WT background, and it restored the almost complete lack of GCs to WT levels in the *BRCA1*$^{-/-}$ background (Fig. 4b). The inactivation of *KU70*

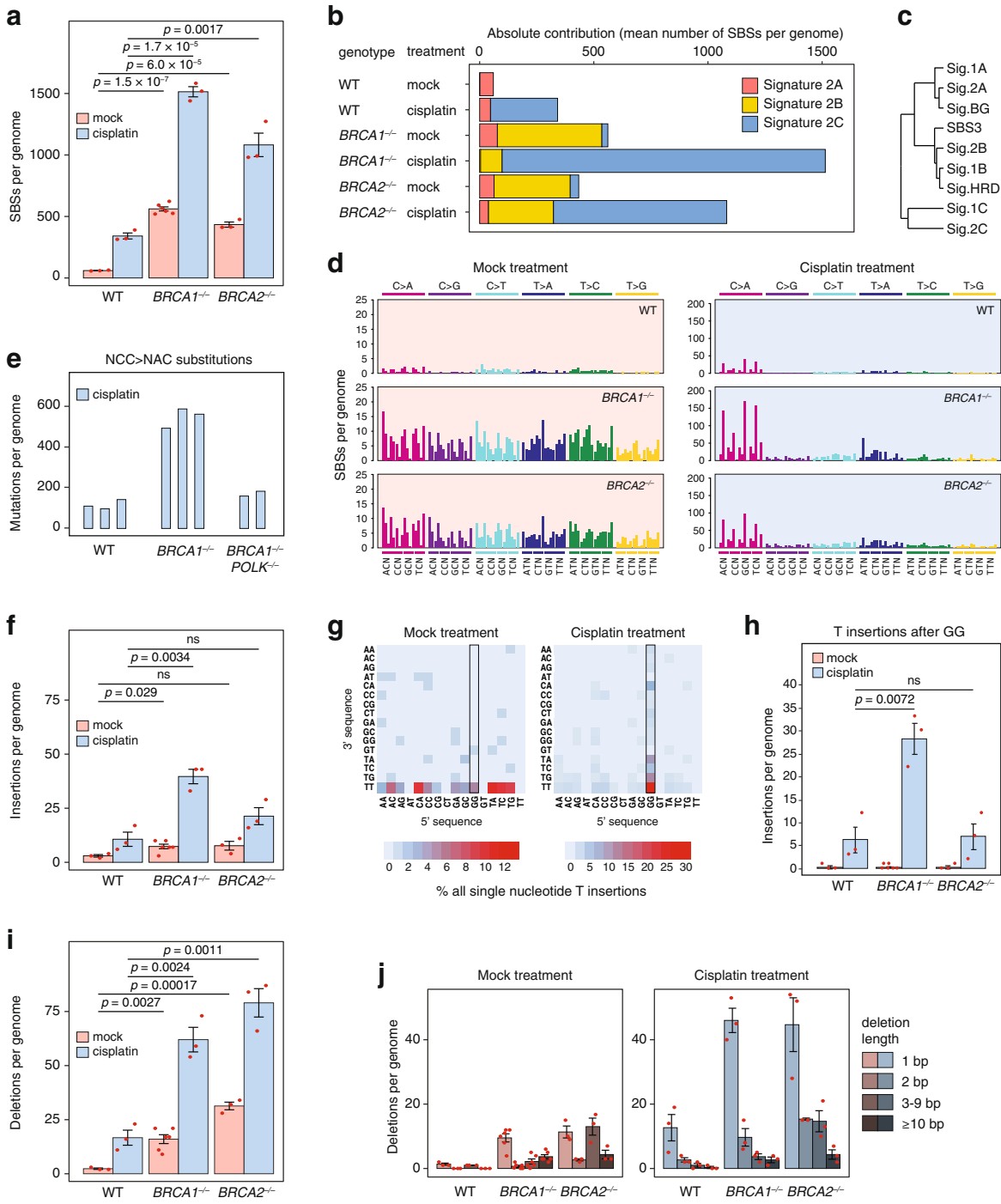

**Fig. 2 Increased TLS-mediated mutagenesis at replication-stalling lesions in BRCA1/2 deficient cells. a** The mean number of SBS (single base substitution) mutations detected in DT40 cell clones of the indicated genotypes following mock treatment or four 1-h treatments at weekly intervals with 10 μM cisplatin. **b** Deconstruction of the averaged triplet SBS spectra of the indicated genotypes and treatments into three mutational signatures using non-negative matrix factorisation. **c** Hierarchical clustering of newly defined and published SBS signatures, see text for details. **d** Mean triplet spectra of the SBS mutations shown in panel (**a**), presented as in Fig. 1e. Note the different *y* axis scale used for mock and cisplatin treatments. **e** The total number of NCC > NAC substitutions in individual sequenced cisplatin-treated samples of the indicated genotypes. **f** The mean number of insertion mutations in the samples presented in (**a**). **g** The sequence context of single T insertions, presented as a heat map of summed events in all mock treated (left panel) or cisplatin-treated (right panel) samples shown in (**f**). **h** The mean genomic number of T insertions after GG dinucleotides (boxed in panel **g**) in samples of the indicated genotype and treatment. **i** The mean number of deletion mutations in the samples presented in (**a**). **j** The mean number of deletions presented in categories separated according to deletion size. Error bars indicate SEM on all panels, individual values are also shown with red symbols on (**a**, **f**, **h**–**j**). The significance of tested differences is shown (ns not significant, unpaired two-sided *t*-test, data presented in (**a**, **f**, **h**–**j**) were derived from genome sequencing of n = 3 biologically independent samples per genotype and treatment except n = 6 for mock treated *BRCA1⁻/⁻* cells). Source data are provided as a Source Data file.

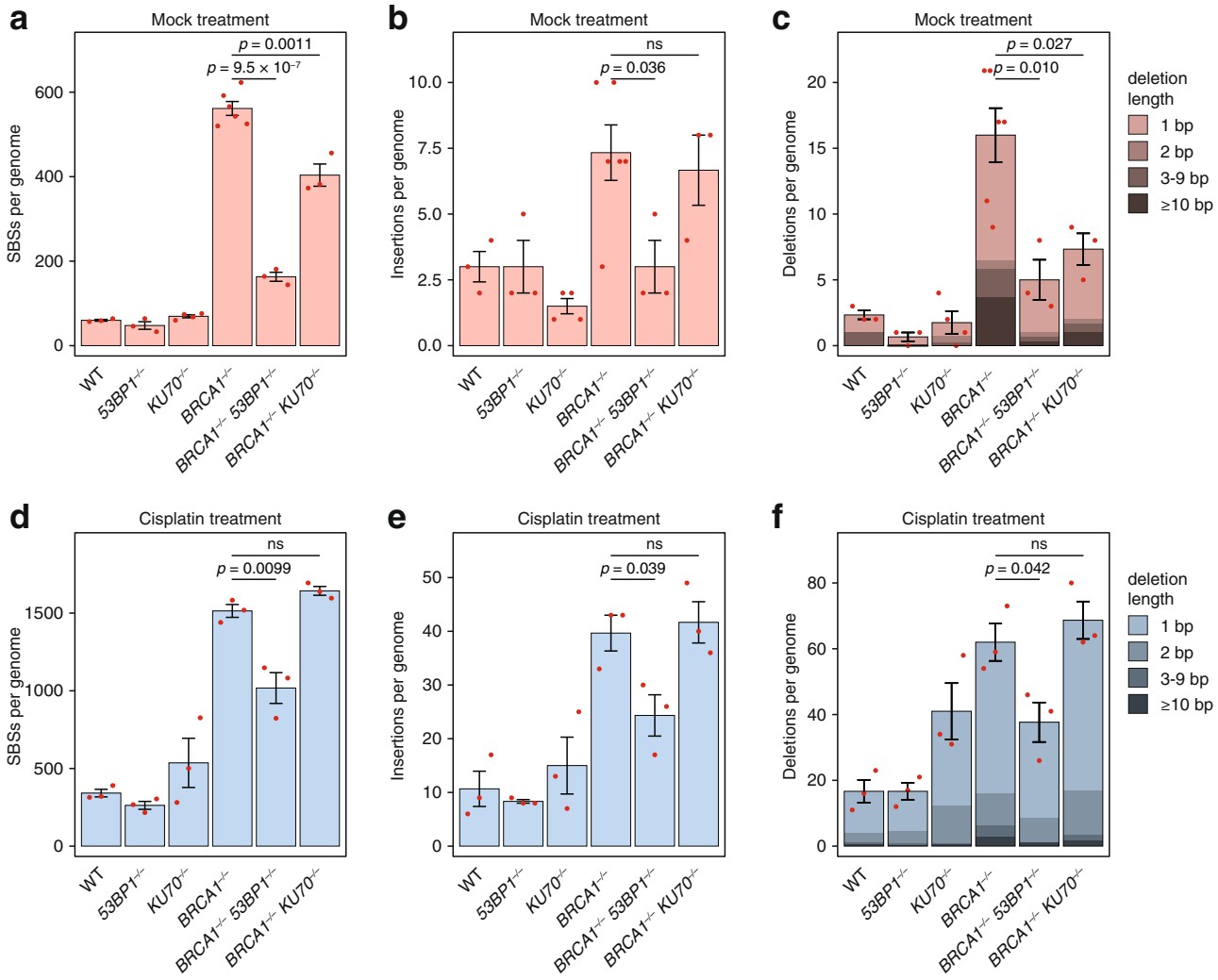

**Fig. 3 Mutagenesis in BRCA1 deficient cells is regulated by 53BP1. a-c** The mean number of newly arising SBS (single base substitution, **a**), insertion (**b**) and deletion (**c**) mutations detected in DT40 cell clones of the indicated genotypes following 50 days of culturing an ancestral clone (mock treatment). **d-e** The mean number of newly arising SBS (**d**), insertion (**e**) and deletion (**f**) mutations detected in DT40 cell clones of the indicated genotypes following 50 days of culturing that included four 1-h treatments at weekly intervals with 10 μM cisplatin. A sub-classification of deletion events according to size is shown by shading in (**c**, **f**). Error bars indicate SEM of the column totals, individual values are also shown with red symbols. The significance of tested differences is shown (ns not significant, unpaired two-sided *t*-test, data were derived from genome sequencing of biologically independent samples, see Supplementary Data 4 for sample sizes). Source data are provided as a Source Data file.

did not affect mutagenesis, confirming that the effect of 53BP1 in suppressing template switching is independent of its function in controlling NHEJ.

We also catalogued the lengths of GC events, recording the maximum possible sequence length that could have been copied from any pseudogene. In addition to being reduced in number, GC tracts in the *BRCA1*⁻/⁻ mutant were significantly shorter than those in the WT (Fig. 4c). The deletion of *53BP1* did not alter GC lengths in the WT background; and restored the mean GC length to the WT level (from 18 nt to 29 nt) in the *BRCA1*⁻/⁻ genetic background (Fig. 4c). A plausible explanation is that 53BP1 inhibits the exonucleolytic enlargement of single-stranded DNA (ssDNA) regions in the absence of BRCA1 to inhibit GC formation, and to reduce the length of the few successful GC tracts.

**BRCA1 and 53BP1 both contribute to tolerance to DNA adducts.** In order to better understand the relative roles of BRCA1 and 53BP1 in DNA damage tolerance, we used cytotoxicity assays

to measure the sensitivity of the single and double mutant DT40 cell lines to a range of cytotoxic agents directed against DNA and its metabolism. *53BP1*⁻/⁻ cells were moderately hypersensitive to the adduct-forming agents cisplatin and methyl methanesulfonate (MMS), whereas *KU70*⁻/⁻ cells were not, supporting the specific NHEJ-independent role of 53BP1 in damage tolerance (Fig. 5a, d). *BRCA1*⁻/⁻ *53BP1*⁻/⁻ cells were more sensitive to cisplatin and MMS than either single mutant, suggesting that the function of 53BP1 in damage bypass goes beyond antagonising BRCA1-dependent HR. Inhibitors of topoisomerase I and PARP-1 both cause the accumulation of DNA single-strand breaks, which can cause replication fork collapse[44,45]. As expected, only the *BRCA1*⁻/⁻ single mutants showed hypersensitivity to such agents, the topoisomerase I inhibitor SN-38 and the PARP inhibitor olaparib (Fig. 5b, e). The disruption of *53BP1* decreased the olaparib sensitivity of *BRCA1*⁻/⁻ cells more than threefold in concordance with earlier findings[17,18], showing that 53BP1 retains its function in regulating DNA break repair in this experimental model. Finally, we tested two inhibitors of topoisomerase II, etoposide and the anthracycline daunomycin, which cause

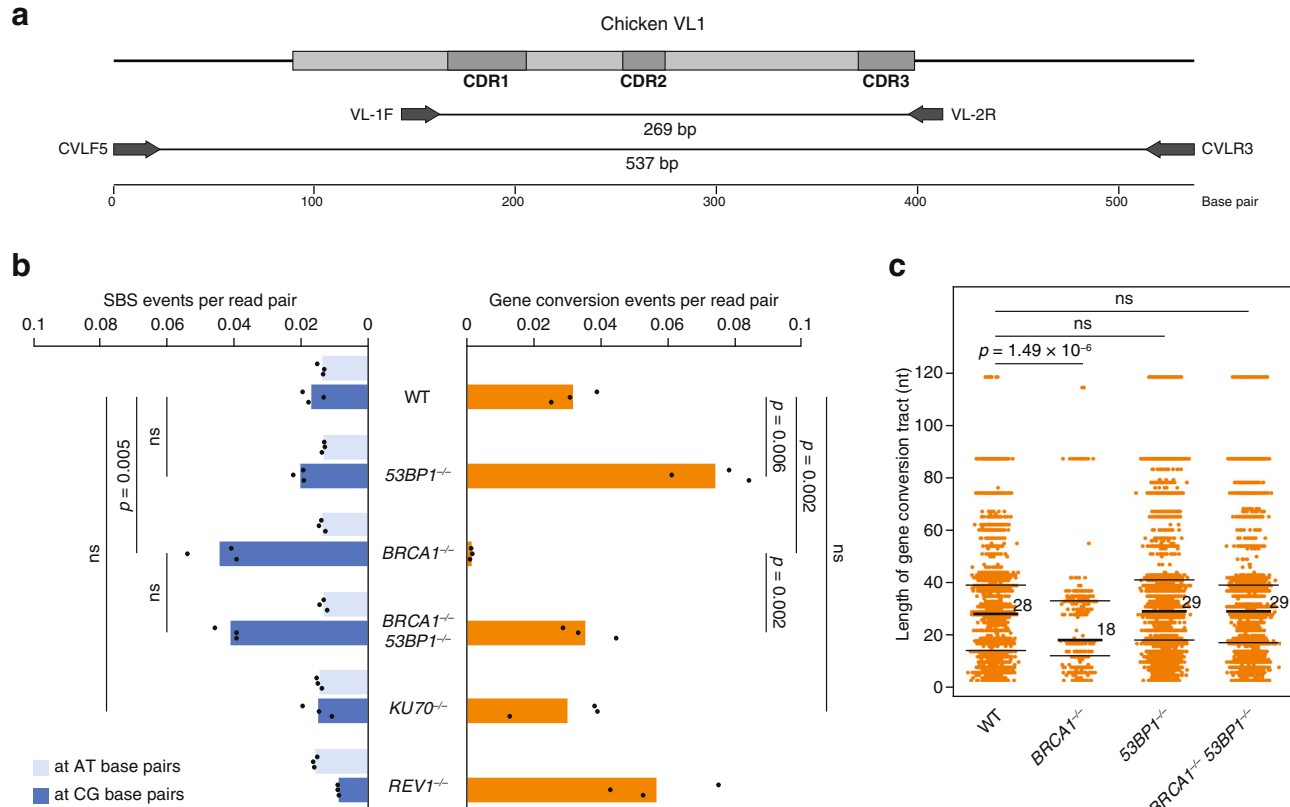

**Fig. 4 53BP1 suppresses gene conversion in the absence of BRCA1. a** Schematic map of the rearranged chicken immunoglobulin VL1 gene. The coding region is shown in grey, darker grey boxes indicate the hypervariable complementarity-determining regions (CDR). Primer pairs for an initial genomic PCR reaction (CVLF5 and CVLR3) and a subsequent nested indexed PCR (VL-1F and VL-2R) are shown. **b** SBS and gene conversion events detected in cell populations derived from individual DT40 cell clones of the indicated genotypes that were cultured for 40 days, presented as the number of mutation events per total number of sequenced read pairs in the same sample. Detected point mutation (SBS) events at AT or CG base pairs are shown separately. Bars show means, individual values of 3 parallel cell clones are indicated with black markers. The significance of tested differences is shown (unpaired two-sided *t*-test, ns not significant). **c** The maximum possible length of each individual gene conversion tract in sequence reads derived from cell populations of the indicated genotypes. The median, upper and lower quartiles are shown, the indicated statistical differences were calculated with the Wilcoxon rank-sum test. Source data are provided as a Source Data file.

replication-independent DSBs[46]. We only observed hypersensitivity in cells that were mutant for *KU70* alone or in combination with *BRCA1*, indicating the integral role of KU70 in DNA end joining that is not shared by 53BP1 (Fig. 5c, f).

**53BP1 promotes TLS in vitro**. The decreased spontaneous and damage-induced mutagenesis and increased DNA damage sensitivity seen in *53BP1* mutant cells suggests a direct role for the protein in promoting error-prone DNA damage bypass. To explore this further, we made use of an in vitro damage bypass assay that relies on DNA replication driven by the SV40 large T antigen. We have recently established and characterised a variant of this assay, whereby DNA lesions are placed on each strand of a shuttle plasmid in a staggered arrangement opposite mismatched bases, and the DpnI-resistant unmethylated replication products are quantified using sequence-specific quantitative PCR. Human cytosolic extract can support TLS over T–T cyclobutyl pyrimidine dimer (CPD) ultraviolet lesions[47], typically producing AA insertions opposite the lesion (Fig. 6a). Template switching would result in GC sequences at both lesion, but we found no strong evidence of such activity in the system[48]. Instead, GC/GC outcomes are primarily the product of active nucleotide excision repair (NER). Another sequence outcome is the appearance of short deletions at the site of either lesion (Fig. 6a), which are

produced by the PCR polymerase and correlate with incomplete replication, likely the presence of CPD lesions in ssDNA gaps surrounded by replicated DNA[48,49]. We tested in vitro CPD lesion bypass in extracts prepared from WT and *53BP1*[−/−] human TK6 lymphoblast cells[50]. Lesion-containing and lesion-free templates replicate with equal efficiency in this assay[48]. The efficiency of plasmid replication, measured by qPCR as the ratio of DpnI-resistant to total recovered plasmid, was significantly lower in cytosolic extract from *53BP1*[−/−] cells than from WT cells (p = 0.002, t-test; Fig. 6c). Although the cytosolic extract contains both cytoplasmic and soluble nuclear proteins, not even the WT extract contained detectable amounts of 53BP1 (Fig. 6b). We therefore supplemented the reactions with two different amounts of nuclear extract from the same cell type, which contained chromatin-bound proteins including 53BP1 in case of the WT cells (Fig. 6b). Supplementation with 4 μg nuclear extract preserved the significant difference in replication efficiency between WT and *53BP1*[−/−] cells (p = 0.002), whereas 20 μg nuclear extract reduced replication efficiency in both genetic backgrounds (Fig. 6c). TLS was the dominant sequence outcome amongst replicated plasmids in all reactions (Fig. 6d). However, TLS-specific replication products were reduced in reactions that contained extracts from *53BP1*[−/−] cells as compared to WT cells, and the difference was significant in case of the reactions supplemented with 53BP1-rich nuclear extract (p = 0.0005 and

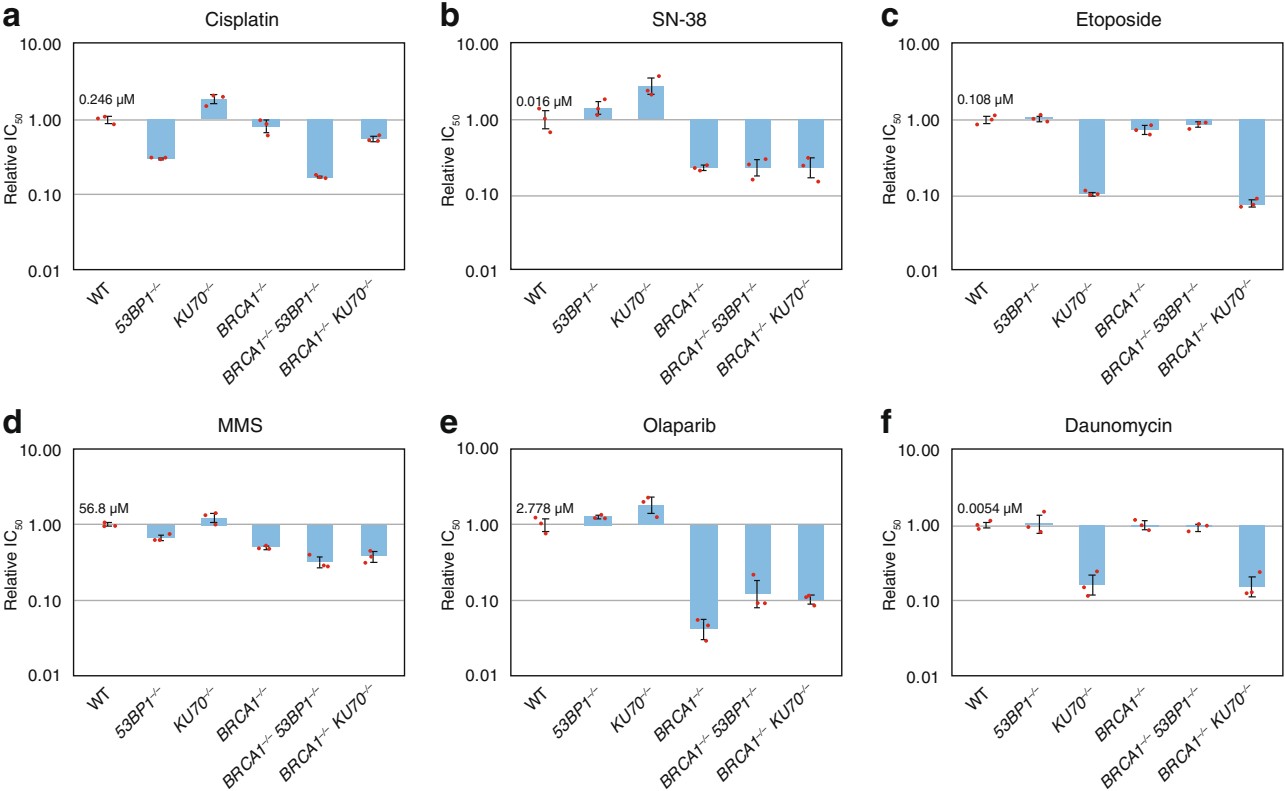

**Fig. 5 Interaction of BRCA1 and 53BP1 in the tolerance to cytotoxic agents. a–f** Cytotoxicity measurements on DT40 cell lines of the indicated genotypes using drugs named above each panel. Results are shown as mean and SEM of relative IC$_{50}$ concentrations compared to the WT cell lines, individual values derived from $n = 3$ independent experiments are indicated. The mean IC$_{50}$ value of the WT cell line is displayed on each panel. Source data are provided as a Source Data file.

$p = 0.010$ in case of 4 µg and 20 µg nuclear extract, respectively; Fig. 6d). The decrease in TLS was accompanied by an increase of NER/template switch–specific outcomes, and we have previously shown that these products are primarily due to NER[48]. We attempted to reconstitute the reactions which contained 4 µg nuclear extract with 53BP1 protein. Lacking a full-length recombinant version of this large disordered protein, we used a fragment encompassing residues 1053-1711 which is sufficient for the formation of nuclear foci at sites of DNA damage[51]. This fragment partially rescued the defects observed in the extracts from *53BP1*$^{-/-}$ cells, significantly elevating both replication efficiency and the proportion of TLS outcomes (Fig. 6c, d). 53BP1 therefore appears to directly promote TLS, especially in the presence of a high concentration of nuclear proteins.

## Discussion

The presented study explored the mechanisms of base substitution and short indel mutagenesis in *BRCA1* deficient cells. We showed that loss of individual TLS polymerases reduces the spontaneous formation of SBSs and indels, and loss of Polκ also changes the mutation spectrum of *BRCA1*$^{-/-}$ cells. A much greater number of SBS and indel mutations at sites of putative cisplatin-induced intrastrand crosslinks further supported our conclusion that an increased use of error-prone TLS is responsible for the majority of mutagenesis in *BRCA1/2* deficient cells. Using both whole-genome sequencing and the analysis of somatic hypermutation at the chicken immunoglobulin Vλ locus, we demonstrated a previously unrecognised function for 53BP1 in regulating the choice between error-prone TLS and error-free HR-dependent template switching lesion bypass. Our results

contribute to understanding the evolution of *BRCA1/2* deficient cancers.

*BRCA1* mutant cells develop mutations of a range of distinct classes, of which an explanation for base substitution mutagenesis has been the most elusive. The high number and generally non-clustered positioning of spontaneous SBS mutations makes it improbable that they form in the process of DSB repair by HR. Also, SBS mutagenesis is indistinguishable in cells lacking BRCA1, BRCA2, RAD51 paralogues or accessory proteins, whereas there are important differences in the formation of deletions with microhomology[8]. DSB resection is hampered in *BRCA1*$^{-/-}$ cells but not in *BRCA2* mutants, thus the formation of microhomology deletions but not of SBS mutations correlates with functional 5′ strand resection. This also argues against processes associated with break repair being the cause for HRD spectrum SBS mutagenesis. Instead, the decrease of spontaneous SBS mutagenesis upon the inactivation of TLS polymerase genes in *BRCA1*$^{-/-}$ cells pointed to the role of TLS. Our results fully agree with observations that increased mutagenesis in *rad51* or *rad52* mutant yeast cells was suppressed by knocking out *rev1* or *rev3*[52,53]. The disruption of *POLH*, *POLK* or *REV1* in *BRCA1*$^{-/-}$ cells did not restore the WT level of mutagenesis, which could be due to redundancy between TLS polymerases. The primary function of REV1 in TLS is as a scaffold protein that aids the recruitment of further polymerases[25], therefore the precise role of each TLS factor in mutagenesis connected to BRCA loss will need further investigation. However, the finding that the deletion of the *POLK* gene in *BRCA1*$^{-/-}$ cells changes the spontaneous SBS spectrum and causes the appearance of sequence-specific SBS peaks further suggests TLS involvement in the remaining SBS mutagenesis. The strongest evidence for the role of TLS in HRD

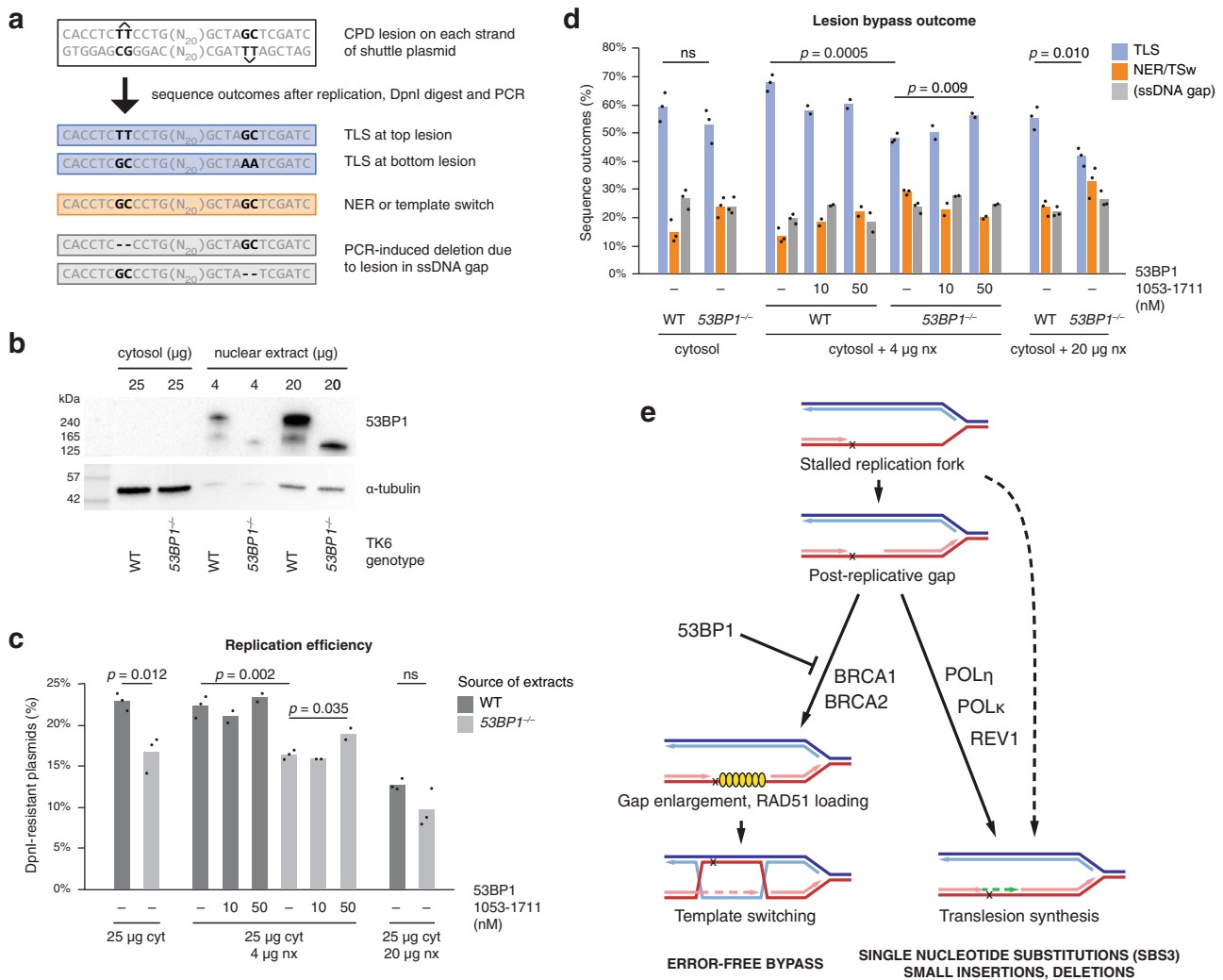

**Fig. 6 53BP1 promotes TLS in human cell extracts. a** Sequence detail of the lesion-containing shuttle plasmid used in in vitro DNA replication reactions in the presence of SV40 T antigen, indicating the staggered position of TT(CPD) lesions on each strand. Below, typical sequence outcomes detected in PCR products of DpnI-digested replicated DNA are shown (top strand sequence), together with the responsible mechanisms. CPD, cyclobutyl pyrimidine dimer, TLS, translesion synthesis, NER, nucleotide excision repair, ssDNA, single-stranded DNA. **b** Western blot of 53BP1 in samples of cytosolic and nuclear extract from WT or *53BP1*−/− human TK6 cells as used in subsequent experiments. A representative image of three experiments is shown. **c** The efficiency of replication, as measured by the ratio of quantitative PCR products spanning 3 or 0 DpnI sites on the lesion-containing plasmids recovered from 10 μl reactions that contained the indicated amounts of cytosolic (cyt) or nuclear extracts (nx) from WT or *53BP1*−/− TK6 cells and the indicated concentrations of a recombinant 53BP1 fragment. **d** Sequence outcomes in the replication reactions also shown in panel (**c**), measured using sequence-specific quantitative PCR on longer PCR products derived from DpnI-resistant replicated lesion-containing plasmids which were incubated in the indicated extracts. TSw, template switching. Means and individual values of independent experiments are shown in (**c, d**), the significance of tested differences is indicated (ns not significant, unpaired two-sided *t*-test, n = 3 independent experiments except n = 2 for the reconstitution experiments). **e** A schematic model of the choice between DNA damage bypass mechanisms that influence mutagenesis in the presence or absence of BRCA1/2. Yellow symbols represent RAD51. Source data are provided as a Source Data file.

mutagenesis came from the treatment of *BRCA1*−/− and *BRCA2*−/− cells with cisplatin, which induced extra mutations with a spectrum that is tied to the putative lesion sites, but did not induce mutations with an HRD spectrum. Indeed, the cisplatin-induced extra mutations in *BRCA1*−/− proved to be dependent on Polκ. From this, we can conclude that SBS mutations in HRD cells arise from lesion-specific processes; and DNA adducts primarily cause mutations via TLS in *BRCA1/2* deficient cells, therefore spontaneous HRD spectrum mutations are also likely to be caused by TLS. Increased TLS activity in BRCA1 deficient cells is also reflected by the reported increase in the recruitment of Polη to subnuclear foci[54]. Importantly, our results also demonstrated that a significant proportion of short insertions and

deletions are caused by TLS in *BRCA1*−/− cells, presumably by frameshift mechanisms at sites of lesions[55,56].

The increased use of TLS in HR mutants is best explained by a role for BRCA1 and other HR factors in an error-free mechanism for which TLS provides a back-up, i.e. a role in replicative lesion bypass. The involvement of HR factors in DNA damage bypass has been described in yeast, and may take place by post-replicative template switching, or by the HR-mediated salvage of a collapsed and broken replication fork[27,57]. Although poorly understood in higher eukaryotes, template switching was shown to take place at post-replicative ssDNA gaps rather than DNA ends in yeast[58], and TLS could indeed provide an alternative to template switch-mediated gap filling. In support of this, TLS and

Rad51-dependent HR have been shown to be redundant alternatives in the processing of nucleotide excision repair–generated gaps in non-dividing G2 phase yeast cells[59]. The generation of post-replicative gaps via PrimPol-mediated re-priming has been shown to promote RAD51 loading at bulky lesions in human cells[60], and RAD51 promotes the repair of ssDNA gaps behind replication forks[61]. BRCA1/2-mediated template switching and TLS are therefore two alternative solutions for the filling of post-replicative gaps (Fig. 6e), and the nearly tenfold higher rate of SBS mutagenesis in DT40 cells deficient for BRCA1, BRCA2 or other HR factors[7,8] suggests that error-free recombinational bypass is the preferred route in WT cells.

The discovered function for 53BP1 in suppressing template switching also sheds light on the potential roles of BRCA1. The observed greater hypersensitivity of $53BP1^{-/-}$ cells to replication-stalling adducts than to DSBs suggests that the regulation of bypass pathway choice may be the primary function of 53BP1. At DNA DSBs 53BP1 separately inhibits 5′-3′ end resection and RAD51 loading[62], and it is plausible that 53BP1 performs analogous functions at post-replicative gaps, where the nucleolytic enlargement of the gaps is required for the loading of RAD51[60]. BRCA1 may antagonise 53BP1 at either step (Fig. 6e). BRCA1 is likely to regulate the enlargement of the ssDNA gaps, as shorter immunoglobulin GCs were generated in its absence. However, its function in promoting RAD51 loading must also be critical in reducing mutagenic TLS activity, as the RAD51 paralogue genes RAD51C, XRCC2 and XRCC3 showed identical SBS phenotypes to BRCA1[8], and the primary function of the encoded proteins is the assembly of the RAD51 nucleoprotein filament[63]. RAD51 loading at reversed forks is also promoted by BRCA1/2 and the RAD51 paralogs, but this is unlikely to influence mutagenesis, as 53BP1 plays no role in the protection of reversed forks[23]. It is an open question how BRCA1 and 53BP1 antagonise each other at replication-stalling lesions, and whether their interacting factors known for regulating DSB repair pathway choice also play a role in DNA damage bypass.

The contribution of HRD-related SBS and short indel mutagenesis to tumorigenesis also requires further investigation. Cancer susceptibility in BRCA1 and BRCA2 mutation carriers has generally been attributed to increased chromosomal instability in BRCA1/2 deficient cells[64]. We did not address the formation of large structural genomic variations in this work, as few rearrangements form in the duration of these experiments in BRCA1 deficient cells[7], thus whole-genome sequencing is an inefficient method for assaying large-scale rearrangements. However, BRCA1 and BRCA2 mutant cells or cancers have substantially different structural rearrangement spectra, whereas their SBS phenotypes are indistinguishable[5,8], which suggests that increased SBS mutagenesis has an important role in the similar tumorigenic consequences of BRCA1 or BRCA2 gene defects.

## Methods

**Cell culture and drug treatments**. The following DT40 cell lines were used: WT, $BRCA1^{-/-}$, $BRCA2^{-/-}$, $POLH^{-/-}$, $POLK^{-/-}$, $REV1^{-/-}$, $KU70^{-/-}$, $53BP1^{-/-}$ (see Supplementary Table 1 for sources). The $BRCA1^{-/-}$ $POLH^{-/-}$, $BRCA1^{-/-}$ $POLK^{-/-}$, $BRCA1^{-/-}$ $REV1^{-/-}$, $BRCA1^{-/-}$ $KU70^{-/-}$, $BRCA1^{-/-}$ $53BP1^{-/-}$ cell lines were generated by disrupting the BRCA1 gene using homologous gene targeting[65] in the respective single mutant lines. All new cell lines are available from the authors. All cell lines and gene mutations were verified using the whole-genome sequence data.

Cells were grown at 37 °C under 5% $CO_2$ in Roswell Park Memorial Institute-1640 medium supplemented with 7% foetal bovine serum, 3% chicken serum and 50 μM 2-mercaptoethanol. Suppliers and catalogue numbers are listed in Supplementary Table 3.

For the whole-genome sequencing, a single cell ancestral clone was isolated from each cell line, expanded and cultured for 50 days. Where indicated, cisplatin treatments were performed after day 20 in four rounds in weekly intervals; one million cells were treated each time for one hour. At the end of the experiment, single-cell clones were isolated and expanded to two million cells prior to genomic DNA preparation using the Gentra Puregene Cell Kit (Qiagen).

**Sensitivity measurements**. For cytotoxicity assays, 1000 DT40 cells per well in 384-well plates were incubated with cytotoxic drugs (Supplementary Table 3) at a range of concentrations using threefold (fourfold in the case of olaparib) dilution series. Cell viability was measured after 72 h using PrestoBlue (Thermo Fisher) and an EnSpire plate reader (Perkin-Elmer). Three technical replicates were averaged per experiment. Data were normalised to untreated cells; curves were fitted with the GraphPad Prism software using the sigmoidal dose–response model. Curve fit statistics were used to determine $IC_{50}$ values.

**Whole-genome sequencing, mutation calling and data analysis**. Library preparation and DNA sequencing were done at the Research Technology Support Facility of Michigan State University, USA (8 samples), and at Novogene, Beijing, China (66 samples) on Illumina HiSeq instruments in 2 × 125 bp or 2 × 150 bp paired-end format, or at BGI, Hong Kong, China using 2 × 150 bp paired-end DNBSeq (4 samples), as specified in Supplementary Data 1. Typically three independent clones were sequenced from each treatment. The mock treatment of $BRCA1^{-/-}$ cells was performed twice with the samples subjected to different sequencing methods; the results were analysed together. All datasets from successfully sequenced samples were used for subsequent analysis.

The reads were aligned to the chicken (Gallus gallus) reference sequence Galgal4.73 as described[7]. Independently arising SBSs and short indels were identified in batches with a modified version of IsoMut[30], that uses samtools with the -E flag during pileup generation to enhance sensitivity towards complex events. The output was post-filtered in R, using packages tidyverse 1.3.1, GenomicRanges 1.44.0, BSgenome.Ggallus.UCSC.Galgal4 1.4.0 and superheat 0.1.0, such that no more than five SBSs and one indel were detected in the ancestral clones, as mutations detected as unique in these samples provide an internal control for false positives. Because of sequencing contamination, we filtered the mapping quality of mutation-supporting reads to minimum 40. Furthermore, we also used the Rsamtools 2.8.0R package to detect and filter mutations that had jumps of 5 or more in coverage on both sides in a 200 bp window, as these events typically mark sites of misalignment or interspecies cross-contamination.

Two-sided unpaired t-tests were used for statistical comparisons of mutation numbers. Unadjusted p values are provided in all figures, $p > 0.05$ was considered not significant. Individual SBS spectra were averaged for each genotype (Supplementary Fig. 1). De novo NMF decomposition and fitting of triplet signatures was performed using the R package MutationalPatterns 3.0.1[66]. For de novo NMF on experimental data, an optimal component number of three was chosen based on the cophenetic correlation coefficient values (Supplementary Fig. 3). SBS spectra were normalised to the frequencies of triplet occurrence in the human genome for comparison to COSMIC signatures in Supplementary Fig. 4.

**Human cancer genomic and transcriptomic data and deconstruction of mutational spectra**. Somatic mutation data of human tumours from various tissues were obtained from PCAWG[6]. Signature deconstruction was performed using the deconstructSigs 1.8.0R package[67] using version 3.1 of the COSMIC signatures[6] supplemented with the NMF-derived SBS.POLKΔ signature. For comparisons to COSMIC using Pearson correlation, triplet spectra were adjusted by multiplying with the ratio of triplet occurrences in the human and chicken genome. The transcriptome of human tumours was obtained from the ICGC portal[68]. Unpaired two-sample Wilcoxon tests were used for statistical comparison of SBS.POLKΔ contributions.

**Immunoglobulin hypermutation assay**. For assaying immunoglobulin hypermutation in DT40 cells, we first sorted cells based on surface IgM expression using a Beckton Dickinson FACSAria cell sorter and a population of cells incubated for 1 h in PBS, 1% FBS and anti-chicken IgM-FITC conjugate (1:100, Bethyl Laboratories, A30-102F). We selected 3 IgM-positive single clones per cell line and cultured them for 40 days, followed by lysis and genomic DNA preparation as above. We amplified VL1 using the primers CVLF5 (CAGGAGCTCGGCTCTGTCCCATTGCTGCGCGG) and CVLR3 (GCGCAAGCTTCCCCAGCCTGCCGCCAAGTCCAAG)[69] for 10 cycles using the high-fidelity thermostable polymerase Pfu Turbo (Agilent). We then proceeded to a second round of PCR for amplicon sequencing using barcode tagged primers VL-1F-bc (GCxxxxxxGAGAAACCGTCAAGATCAC) and VL-2R-bc (GCxxxxxxTTGTCCCGGCCCCAAATG), where xxxxxx shows the position of the sequence barcodes. The second PCR amplified a 269 bp amplicon, using which we could detect somatic hypermutation in a 233 bp part of the VL1 coding region that includes variable regions CDR1, CDR2 and CDR3. The purified PCR products were mixed for each cell line, subjected to library preparation with the NEBNext Ultra II DNA Library Prep kit and sequencing on an Illumina MiSeq instrument using 2 × 300 nt paired-end sequencing. Raw sequencing reads were preprocessed first with Trimmomatic[70] to remove Illumina adaptors and sequences of low quality, and with FLASH2[71] to merge overlapping pairs. Pre-expansion amplicons were Sanger sequenced to identify genotype-specific variations. Pseudogene sequences were taken from[72]. Analysis of the alterations in each amplicon sequence was conducted in python 3.8.5, using the Biopython library (version 1.79)[73]. The post-expansion amplicons of different genotypes were individually compared against the respective pre-expansion sequences using global alignment. The detected variants were filtered for base qualities more than 30, and grouped into three categories: point mutation if the variant was not present on any of the pseudogenes, GC if at least two variants of

the same amplicon were concurrently present on the same pseudogene and their distance was at most 9 bps, or ambiguous if the two variants were present on different pseudogenes or their distance was greater than 9 bps. GC tract lengths were determined by measuring the distance between the two closest pseudogene-specific variants that were not present on the respective amplicon, and if more pseudogenes were possible, the shortest tract length was chosen.

**Recombinant protein purification.** The 53BP1 fragment encompassing residues 1053-1711 was expressed in *Escherichia coli* BL21-CodonPlus (DE3)-RP upon induction with 300 μM isopropyl-ß-D-1-thiogalactopyranoside for 16 h at 17 °C. Harvested cells were resuspended in lysis buffer [50 mM Tris·HCl pH 7.4, 50 mM NaCl, 5% glycerol, 2 mM DTT, 0.1% Triton X-100] supplemented with protease inhibitors, stirred for 1 h at 4 °C in the presence of Benzonase Nuclease, and then disrupted by sonication. The clarified supernatant was applied to a Pierce Glutathione Agarose column (Thermofisher) and washed extensively with high-salt and low-salt wash buffer. On-column cleavage of the GST tag was performed overnight using PreScission protease. The cleaved protein was applied on a Resource Q chromatography column (Cytiva). The protein was eluted using a salt gradient, concentrated and then applied on a Superdex 200 Increase 10/300 GL (Cytiva) pre-equilibrated in SEC buffer [50 mM Tris·HCl pH 7.4, 150 mM NaCl, 5% glycerol, 1 mM DTT]. Protein-containing fractions were concentrated and stored in SEC buffer. Suppliers and catalogue numbers are listed in Supplementary Table 3.

**In vitro replication assays.** In vitro replication assays were performed and analysed using recently developed protocols[48]. Thirty nanograms pUCQF(CPD) plasmid containing TT(CPD) ultraviolet lesions on each strand in a staggered arrangement was incubated in cytosolic and nuclear extracts in 10 μl reaction volume in the presence of 0.4 μg SV40 T antigen, 30 mM HEPES (pH 7.6), 7 mM MgCl₂, 0.5 mM DTT, 4 mM ATP, 100 μM each of dATP, dGTP, dTTP, dCTP, 80 μM each of CTP, GTP, UTP, 40 mM phosphocreatine and 0.4 U creatine phosphokinase. Recombinant 53BP1 fragment was added at concentrations in the range of the estimated nuclear 53BP1 concentration. After 4 h, reactions were terminated by the addition of SDS/EDTA to 0.75% (w/v)/22.5 mM final concentrations. DNA was deproteinised by incubation with proteinase K (1 μl of 10 mg/ml) at 55 °C for an hour. Digested proteins were precipitated with ammonium acetate (1.25 M) at 4 °C and centrifuged (20,000 × *g* for 10 min). DNA was precipitated using isopropanol/ethanol, dried, and dissolved in the reaction mixture prepared for DpnI digestion and digested at 37 °C for 60 min with 10 U DpnI in NEB4 buffer (New England Biolabs) supplemented with NaCl to 200 mM final concentration. Replication efficiency was measured as the product ratio of quantitative PCR on regions of the plasmid with 0 or 3 DpnI sites. To determine the sequence outcomes, we used sequence-specific quantitative PCR reactions that were validated against amplicon sequencing results[48]. Quantitative PCR reactions were performed using a Bio-Rad CFX96 instrument and Xceed SG Mix and analysed using Bio-Rad CFX Manager 3.0 software. All primers are listed in Supplementary Table 2; suppliers and catalogue numbers of reagents are listed in Supplementary Table 3. In a modification to the published protocol, cytosolic extracts were prepared with the addition of 100 mM NaCl after breaking the cells in hypotonic buffer. The following antibodies were used on Western blots of the extracts: anti-53BP1 (1:500, sc-517281, Santa Cruz Biotechnology), anti-α-tubulin (1:2000, T6199, Sigma-Aldrich). Blots were scanned and analysed using Bio-Rad Image Lab 4.1.

**Reporting summary.** Further information on research design is available in the Nature Research Reporting Summary linked to this article.

## Data availability

The data that support this study are available from the corresponding authors upon reasonable request. Raw whole-genome sequence data generated in this study are available from the European Nucleotide Archive under study accession number PRJEB44196. Mutation lists and further source data for mutational analyses are provided as Supplementary Data. Source data are provided with this paper.

## Code availability

Custom code used in data analysis is available at https://github.com/szutsgroup/BRCA1_TLS_mutagenesis[73].

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

## Acknowledgements
This work was supported by the National Research, Technology and Innovation Fund of Hungary (K_124881, FIEK_16-1-2016-0005 and VEKOP-2.3.3-15-2017-00014 to D.S., NAP2-2017-1.2.1-NKP-0002 to Z.Sza.), the Breast Cancer Research Foundation (BCRF-20-137 to A.L.R. and DS, BCRF-20-159 to Z.Sza.), Kræftens Bekæmpelse (R281-A16566 to Z.Sza.), the Novo Nordisk Foundation Interdisciplinary Synergy Programme Grant (NNF15OC0016584 to Z.Sza.), Department of Defense through the Prostate Cancer Research Program (W81XWH-18-2-0056 to Z.Sza.), Det Frie Forskningsråd Sundhed og Sygdom (7016-00345B to Z.Sza.), and the Basser Foundation (to Z.Sza.). A.L.R. is supported by the Peter and Judy Kovler Professorship in Breast Cancer Research. F.d'A.d.F laboratory is supported by: ERC advanced grant (TELORNAGING—835103); AIRC-IG (21762); Telethon (GGP17111); AIRC 5×1000 (21091); ERC PoC grant (FIREQUENCER—875139); Progetti di Ricerca di Interesse Nazionale (PRIN) 2015 "ATR and ATM-mediated control of chromosome integrity and cell plasticity"; Progetti di Ricerca di Interesse Nazionale (PRIN) 2017 "RNA and genome Instability"; Progetto AriSLA 2021 "DDR & ALS"; POR FESR 2014-2020 Regione Lombardia (InterSLA project); FRRB - Fondazione Regionale per la Ricerca Biomedica - under the frame of EJP RD, the European Joint Programme on Rare Diseases with funding from the European Union's Horizon 2020 research and innovation programme under the EJP RD COFUND-EJP No 825575.

## Author contributions
D.S., A.L.R. and Z.Sza conceived the study; D.C., J.Z.G., Z.Sze, B.S., Z.G., J.Z., M.C. performed the experiments; J.Z.G., A.P. and E.N. performed bioinformatic analyses; D.S. wrote the manuscript; all authors including F.d'AdF analysed data and approved the manuscript.

## Competing interests
The authors declare no competing interests.

## Additional information

**Peer Review Information** *Nature Communications* thanks the anonymous, reviewer(s) for their contribution to the peer review of this work. Peer reviewer reports are available

