## [Peer Review File · Nature Communications]

REVIEWER COMMENTS

Reviewer #1 (Remarks to the Author):

In this study the authors analyze the role of TLS polymerases (Poln, Polk and Rev1) and 53BP1 in spontaneous and DNA-damage induced (cisplatin) mutagenesis (single base substitutions, SBSs and insertions/deletions), in BRCA 1 and BRCA2 defective DT40 cells. They also analyze the role of 53BP1 in gene conversion and somatic hyper-mutation in BRCA1 defective cells at the highly variable gene encoding for the light chain of immunoglobulins. The analysis of the in vitro role of 53BP1 in lesion bypass of a pyrimidine dimer in a replication assay is also presented.

Findings:

1) Poln and Rev1 are responsible for a subset of spontaneous SBSs in BRCA1 defective cells while Polk causes a subfraction of spontaneous insertions. Poln, PolK and Rev1 do not contribute to spontaneous deletions occurring in the absence of BRCA1. Absence of Polk induces the insurgence of a specific SBS signature (signature c) in BRCA1 defective cells.

2) Treatment of BRCA1 and BRCA2 defective cells with cisplatin induces SBS and insertion/deletion mutations at genome sequences that can be (theoretically) susceptible of accumulating intra-strand cisplatin-dependent DNA adducts. BRCA1 and BRCA2 defective cells treated with cisplatin accumulate a specific damage-dependent SBS signature (signature C).

3) Knock out of 53BP1 reduces spontaneous and damage-induced SBS and insertion/deletion mutations in BRCA1 defective cells.

4) Knock out of 53BP1 increases gene conversion at the variable gene encoding for the light chain immunoglobulin gene and its absence in BRCA1 defective cells rescues gene conversion. Absence of 53BP1 does not influence somatic hyper-mutation.

5) 53BP1 knock out leads to sensitivity to DNA replication blocking agent like cisplatin and MMS.

6) 53BP1 is required for the by-pass of pyrimidine dimers located on a plasmid substrate in an in vitro replication assay.

Comments to the authors.

The study of the cellular mechanisms underlying the mutagenesis in BRCA-defective cells is important as it can contribute to clarify the mechanisms of insurgence of breast cancer clones resistant to the treatment with PARP inhibitors.

The approach utilized (knock out into DT40 cells), is advantageous because mutagenesis can be followed in isogenic mutant cell lines and normalized to the sequence of the genome of their "progenitors".

This manuscript deserves to be published in Nature Communications but, from my point of view, there are certain text modifications that should be done to avoid premature statements that are not fully supported by the data presented and to better describe certain parts of the results.

1) I find the statement in the title a bit strong indeed from the data presented in the manuscript, only a subset of spontaneous SBSs induced by the absence of BRCA1 are Poln and Polk dependent. As it may be noticed in figure 1A in the BRCA1^{-/-} POLH^{-/-} and BRCA1^{-/-} REV1^{-/-} mutant cell lines there are still, roughly, 350 SBSs/genome in both cell lines. I advise the authors to smooth down the statement in the title as only a subset of spontaneous SBSs in BRCA1 defective cells is TLS dependent. Moreover, the genetic dependency of the increased signature C (caused by POLH knock out in BRCA1^{-/-} cells) (Figure 1D) is not known. This last concept should be better clarified to the readers as, in this case,

this is not a Polk-dependent signature, but it is a signature that appears in the absence of Polk. In this regard it is not clear if the signature C in figure 1D is the same signature C reported in Figure 2B. The nomenclature of the signatures should be revisited as this could be confusing to non-specialized readers. In general, I would say that a signature is dependent on a gene if it disappears when the corresponding gene is removed.

2) Figure 2 has a point of weakness that should be better clarified to the readers. From my point of view, if the authors would like to show that cisplatin-induced SBSs and insertion/deletions are TLS dependent in BRCA1 and BRCA2 defective cells they should measure those mutagenesis rates in the BRCA1^{-/-} TLS^{-/-} and BRCA2^{-/-} TLS^{-/-} double knock out cell lines treated with cisplatin. If the authors do not have the data in the double knock out cell lines to include in the manuscript, they should better explain to the readers that this is a hypothesis and that it will be necessary to measure those mutagenesis rates in the double knock out cell lines.

3) Figure 4 shows that, in that specific system, 53BP1 acts as a sort of inhibitor of gene conversion. I agree with the authors when they suggest that inactivation of 53BP1 could re-activate the resection of the ssDNA gaps carrying the abasic sites thus re-activating gene conversion in BRCA1 defective cells. In this context, I do not think that the data presented in this study (or somewhere else) are sufficient to support the idea that 53BP1 regulates somatic hypermutation or that it acts regulating the balance between somatic hyper-mutation and gene conversion in this specific locus of the DT40 genome. Those interpretations should be tuned down until the authors will present clear evidence in support of a role of 53BP1 in regulating somatic hyper-mutation at the highly variable light chain gene of DT40. In particular, the fact that the absence of 53BP1 increases gene conversion does not necessary mean that 53BP1 has a role in regulating the balance between gene conversion and somatic hyper-mutation indeed it seems that there are no roles of 53BP1 in regulating somatic hyper-mutation.

Additional points to be addressed

- 1) In figure 1D/E the 3 signatures should be normalized for the content of triplets in the genome of DT40 cells and compared in terms of cosine similarity with all the SBSs identified in tumors
- 2) Same as in point 1 for the signatures in figure 2B
- 3) In figure 1A,1B, 1C etc. the data should be expressed as a mutation accumulation rate (SBSs/genome/generation or SBSs/genome/passage) so that there is a reference for researchers carrying out similar studies.

The concept that defects in HR lead to TLS-dependent mutation signatures is not entirely new and the authors should cite the relevant literature, including for example Endo, K., et al. *Genes Genet. Syst.* 82, 35–42 (2007) and Loeillet, S. et al. *Proc Natl Acad Sci U S A* 62, 202011332 (2020)

- 4) Are signatures A, B, and C the same signatures in Figure 1D and 2B? If not please use different names to avoid confusion.

The nomenclature of SBSs signatures should also be carefully revised. For example, signature B (in Fig 1D) is the main component of mutagenesis in BRCA^{-/-} cells and disappears in the absence of Polk. This signature is therefore a "Polk-dependent signature". Signature C was only minimally contributing to mutagenesis in BRCA^{-/-} cells but becomes the main contributor to mutagenesis once Polk is removed, therefore this signature is caused by an unknown polymerase or other process that becomes predominant in the absence of Polk.

Reviewer #2 (Remarks to the Author):

The manuscript by Chen et al is an interesting study of mutagenesis patterns in BRCA1 deficient cells

that suggests a role for 53BP1 in antagonizing template switch recombination and promoting translesion synthesis (TLS). This topic is of broad scientific interest, particularly given the observation that BRCA-deficient cancers often exhibit a base substitution signature (SBS3), whose etiology has been difficult to pinpoint. The authors use engineered DT40 cell lines in conjunction with genomic sequencing analyses to evaluate the effect of gene disruption on mutation patterns in BRCA1 deficient cells. Whole genome sequencing analyses reveal a modest reduction in single base substitutions (SBS), but not indels, after deletion of TLS enzymes in BRCA1 deficient cells. The study goes on to show that increased spontaneous and cisplatin-induced mutagenesis in BRCA1 deficient cells is 53BP1 dependent. This 53BP1 associated effect is independent of Ku70-mediated NHEJ. Figure 4 shows a role for 53BP1 deficiency in restoring partial gene conversion activity induced by AID in a somatic hypermutation locus, but did not validate a role in promoting SBS in this locus. Thus, Figure 6 is vital to the conclusion, yet is underwhelming in showing a direct role in regulating TLS. Major critiques are listed below:

1) The authors should be transparent in their discussion regarding potential limitations of their sequencing method and analyses. Examination of the methods section suggests there was compromise of the sequencing data due to cross-species contamination that required more stringent filters and in some cases manual processing. This may reduce the sensitivity for structural rearrangements, such as tandem duplications and templated insertions that have been reported in BRCA1 deficient cells (e.g. PMID 32680986, 32680986). These technical limitations may have biased their analyses towards easier-to-detect SBS and small indel signatures.

2) Figure 5 interpretation requires 53BP1 reconstitution experiments to establish whether the observed differences in TLS proficiency are due to 53BP1 deficiency. This is particularly true since cytosolic fractions, that lack detectable 53BP1 protein, have distinct TLS activity.

3) The authors chose to focus their studies on BRCA1 rather than BRCA2. Did they examine the phenotype of BRCA2/53BP1 double knockout cells? This may help to clarify whether the role of 53BP1 in promoting TLS is directly related to or independent from its ability to restore HR-mediated repair.

Reviewer #3 (Remarks to the Author):

The study makes use of gene knockouts in chicken DT40 cells combined with whole genome sequencing to address potential contributing factors to the enhanced level of mutagenesis found in BRCA1 and BRCA2 deficient conditions, in particular in base substitutions. Some of the more interesting findings are i) Translesion synthesis contributes to part of the increased SBS rate, ii) Elevated levels of mutagenesis in HR compromised cells treated with cisplatin, arguing for error-free repair by HR in genetically non-compromised conditions, iii) the observation that 53BP1 apart from restricting gene-conversion in BRCA1 mutant cells (as expected) it also restricts the length of gene-conversion tracts (figure 4c).

I don't consider the study a very good candidate for publication in Nature communication. In my opinion, while I feel that the experiments are solid and well executed, the total is somewhat of a mixed bag, which added up does not provide clear (mechanistic) and conclusive insight into the underlying biology underlying the increased degree of SBS(3) in HR compromised cells. It is largely descriptive, which is not undervalued, but for this type of journal I would expect more substantial advance over what is already known. As an example: the authors show the involvement of TLS, but it would be rather unexpected if not given that TLS is involved in most SBS mutagenesis. The source of the SBS mutagenesis is still unknown, and why it is elevated in HR compromised cells as well.

Apart from this general concern I have note much to comment. As said, I felt that the paper was easy

to read and the experiments well presented. I feel that the study is of interest to experts in the field but less so to a broader audience.

Response to reviewers

We sincerely thank the reviewers for their remarks and suggestions, which led to significant improvements of the manuscript. Below, we provide a point-by-point response to their concerns.

Reviewer #1 (Remarks to the Author):

In this study the authors analyze the role of TLS polymerases (Poln, Polk and Rev1) and 53BP1 in spontaneous and DNA-damage induced (cisplatin) mutagenesis (single base substitutions, SBSs and insertions/deletions), in BRCA 1 and BRCA2 defective DT40 cells. They also analyze the role of 53BP1 in gene conversion and somatic hyper-mutation in BRCA1 defective cells at the highly variable gene encoding for the light chain of immunoglobulins. The analysis of the in vitro role of 53BP1 in lesion bypass of a pyrimidine dimer in a replication assay is also presented.

Findings:

- 1) Poln and Rev1 are responsible for a subset of spontaneous SBSs in BRCA1 defective cells while Polk causes a subfraction of spontaneous insertions. Poln, PolK and Rev1 do not contribute to spontaneous deletions occurring in the absence of BRCA1. Absence of Polk induces the insurgence of a specific SBS signature (signature c) in BRCA1 defective cells.*
- 2) Treatment of BRCA1 and BRCA2 defective cells with cisplatin induces SBS and insertion/deletion mutations at genome sequences that can be (theoretically) susceptible of accumulating intra-strand cisplatin-dependent DNA adducts. BRCA1 and BRCA2 defective cells treated with cisplatin accumulate a specific damage-dependent SBS signature (signature C).*
- 3) Knock out of 53BP1 reduces spontaneous and damage-induced SBS and insertion/deletion mutations in BRCA1 defective cells.*
- 4) Knock out of 53BP1 increases gene conversion at the variable gene encoding for the light chain immunoglobulin gene and its absence in BRCA1 defective cells rescues gene conversion. Absence of 53BP1 does not influence somatic hyper-mutation.*
- 5) 53BP1 knock out leads to sensitivity to DNA replication blocking agent like cisplatin and MMS.*
- 6) 53BP1 is required for the by-pass of pyrimidine dimers located on a plasmid substrate in an in vitro replication assay.*

Comments to the authors.

The study of the cellular mechanisms underlying the mutagenesis in BRCA-defective cells is important as it can contribute to clarify the mechanisms of insurgence of breast cancer clones resistant to the treatment with PARP inhibitors.

The approach utilized (knock out into DT40 cells), is advantageous because mutagenesis can be followed in isogenic mutant cell lines and normalized to the sequence of the genome of their “progenitors”.

This manuscript deserves to be published in Nature Communications but, from my point of view, there are certain text modifications that should be done to avoid premature statements that are not fully supported by the data presented and to better describe certain parts of the results.

- 1) I find the statement in the title a bit strong indeed from the data presented in the manuscript, only a subset of spontaneous SBSs induced by the absence of BRCA1 are Poln and Polk dependent.*

As it may be noticed in figure 1A in the BRCA1-/- POLH-/- and BRCA1-/- REV1-/- mutant cell lines there are still, roughly, 350 SBSs/genome in both cell lines. I advise the authors to smooth down the statement in the title as only a subset of spontaneous SBSs in BRCA1 defective cells is TLS dependent. Moreover, the genetic dependency of the increased signature C (caused by POLH knock out in BRCA1 -/- cells) (Figure 1D) is not known. This last concept should be better clarified to the readers as, in this case, this is not a Polk-dependent signature, but it is a signature that appears in the absence of Polk. In this regard it is not clear if the signature C in figure 1D is the same signature C reported in Figure 2B. The nomenclature of the signatures should be revisited as this could be confusing to non-specialized readers. In general, I would say that a signature is dependent on a gene if it disappears when the corresponding gene is removed.

The reviewer correctly remarks that knocking out any single TLS gene did not reduce the rate of spontaneous mutagenesis in the BRCA1-/- background to wild type levels. This does not, however, argue against all extra base substitutions in BRCA1-/- cells being due to TLS, as TLS is not a single linear pathway with several polymerases and at least two separate recruitment mechanisms through PCNA monoubiquitylation and REV1 (this redundancy is discussed in the manuscript). Knocking out several TLS genes together with BRCA1 may have reduced mutagenesis to wild type levels or below if such cells were viable. The experiments in figure 1 therefore indeed do not fully prove the statement in the title. The data in figure 2 show that cisplatin is more mutagenic in BRCA deficient cells, and that ALL the extra mutations are due to TLS. Our newly incorporated experimental results with BRCA1 POLK cells (see below) further support the notion that all extra mutagenesis is caused by TLS. To integrate these data and also follow the reviewer's advice, we propose to change the title to "BRCA1 deficiency specific base substitution mutagenesis is dependent on translesion synthesis and regulated by 53BP1". The reduction of BRCA deficiency related mutagenesis in TLS deficient cells agrees with it being 'dependent' on TLS, and in certain settings we show that it is in fact fully dependent on TLS.

We have now improved the nomenclature of the NMF-derived mutational signatures. Signature 1C appears in POLK mutants, and we agree with the referee that the genetic dependence of the mutations in this signature is not known. We made clear in the text that the signature is dependent on POLK deficiency (not POLK itself) "Signature 1C is dominated by T>A mutations in a (C/G/T)TT context, and it appears to correlate with the inactivation of the POLK gene, as it also appears in the single POLK-/- mutant". In the explanatory name used for the signature (Sig.POLK Δ), delta stands for deficiency or deletion. We clarified this in the revised manuscript.

As a general note about the names of the signatures, we have renamed them 1A, 1B and 1C in Figure 1, and 2A, 2B and 2C in figure 2. Signature 1C is Sig.POLK Δ , but we did not want to give them all meaningful names as this could amount to overinterpretation of the data. However, in the new figure panel 2C and in a new supplementary figure panel S4A we now show that signatures 1A and 1B are similar to each other and to the previously named BG (background) signature, whereas 2A and 2B are similar to each other and to the HRD and SBS3 signatures. The NMF process gives slightly different results on each dataset, therefore e.g. 1B and 2B are not quite the same, but they both describe the HR deficiency specific pattern.

2) *Figure 2 has a point of weakness that should be better clarified to the readers. From my point of view, if the authors would like to show that cisplatin-induced SBSs and insertion/deletions are TLS dependent in BRCA1 and BRCA2 defective cells they should measure those mutagenesis rates in the BRCA1^{-/-} TLS^{-/-} and BRCA2^{-/-} TLS^{-/-} double knock out cell lines treated with cisplatin. If the authors do not have the data in the double knock out cell lines to include in the manuscript, they should better explain to the readers that this is a hypothesis and that it will be necessary to measure those mutagenesis rates in the double knock out cell lines.*

We thank the reviewer for this comment and suggestion. Because polymerase kappa has been reported to perform mutagenic TLS across intrastrand GG cisplatin crosslinks in human cells (Shachar et al., 2009, EMBO J), we treated BRCA1^{-/-} POLK^{-/-} cells with the same cisplatin regimen as used in other experiments reported in Figure 2. As expected, the incremental cisplatin-related mutagenesis seen in BRCA1^{-/-} cells was reduced, or ‘rescued’ in the double mutants. Specifically, the CC>AC peaks of the spectrum were much reduced, almost entirely down to the WT level (new Figure panel. 2e). We added all the relevant data on the two new WGS samples (cisplatin-treated BRCA1^{-/-} POLK^{-/-} cells) to the supplementary tables. We also added a comment on these results to the discussion.

3) *Figure 4 shows that, in that specific system, 53BP1 acts as a sort of inhibitor of gene conversion. I agree with the authors when they suggest that inactivation of 53BP1 could re-activate the resection of the ssDNA gaps carrying the abasic sites thus re-activating gene conversion in BRCA1 defective cells. In this context, I do not think that the data presented in this study (or somewhere else) are sufficient to support the idea that 53BP1 regulates somatic hypermutation or that it acts regulating the balance between somatic hyper-mutation and gene conversion in this specific locus of the DT40 genome. Those interpretations should be tuned down until the authors will present clear evidence in support of a role of 53BP1 in regulating somatic hyper-mutation at the highly variable light chain gene of DT40. In particular, the fact that the absence of 53BP1 increases gene conversion does not necessary mean that 53BP1 has a role in regulating the balance between gene conversion and somatic hyper-mutation indeed it seems that there are no roles of 53BP1 in regulating somatic hyper-mutation.*

We fully agree with the reviewer that in the immunoglobulin locus 53BP1 does not regulate hypermutation, only gene conversions. We were careful to only state the latter: “The disruption of 53BP1 did not significantly affect base substitution mutagenesis, but it significantly increased the number of gene conversions in the WT background, and it restored the almost complete lack of gene conversions to WT levels in the BRCA1^{-/-} background”. As the referee remarks, these results are useful to suggest a mechanism by which 53BP1 regulates replicative damage bypass.

We are not sure why 53BP1 affects TLS in the case of genome-wide spontaneous and cisplatin-induced mutagenesis, but not in the case of abasic site-induced mutagenesis in the Ig locus. The nature of the lesions may account for the difference.

We tuned down the conclusion that 53BP1 regulates the choice between TLS and error-free template switching in replicative DNA damage bypass, and do not claim this in the context of Ig diversification.

Additional points to be addressed

- 1) *In figure 1D/E the 3 signatures should be normalized for the content of triplets in the genome of DT40 cells and compared in terms of cosine similarity with all the SBSs identified in tumors*
- 2) *Same as in point 1 for the signatures in figure 2B*

We have performed this comparison and included the results in a new supplementary figure (Fig. S4B) and a new supplementary table S6.

- 3) *In figure 1A,1B, 1C etc. the data should be expressed as a mutation accumulation rate (SBSs/genome/generation or SBSs/genome/passage) so that there is a reference for researchers carrying out similar studies.*

In figure 1 we changed the axis labels to “SBSs per genome in 50 days” etc. We prefer not to divide the numbers by the 50 days, or the approximately 100 generations that take place in this time to maintain comparability with Figure 2. In Figure 2 we are showing the mutagenic results of cisplatin treatments. As explained in the text and the methods, these treatments are not continuous, but rather take place in four weekly cycles over 50 days starting on day 20, with 1h treatments each time exactly as in our earlier publications. The cisplatin-induced mutations therefore cannot be allocated to individual cell cycles or passages, thus the total mutation numbers must be shown for both these and the matching mock treatments.

We hope the extended axis labels and our addition that 50 days is about 100 generations for DT40 cells is sufficient for comparisons.

The concept that defects in HR lead to TLS-dependent mutation signatures is not entirely new and the authors should cite the relevant literature, including for example Endo, K., et al. Genes Genet. Syst. 82, 35–42 (2007) and Loeillet, S. et al. Proc Natl Acad Sci U S A 62, 202011332 (2020)

We thank the reviewer for the request, and are now citing these studies. We also inserted a new citation of an elegant study in which the competition of HR and TLS was demonstrated in the context of single strand gaps generated by NER in non-replicating yeast cells (Ma et al. 2013, PNAS).

- 4) *Are signatures A, B, and C the same signatures in Figure 1D and 2B? If not please use different names to avoid confusion.*

We apologise for the confusing nomenclature, these signatures are indeed not the same. As mentioned above, we have clarified the names of the newly derived signatures, and provided a supplementary figure that clusters them by similarity and provides similarity values.

The nomenclature of SBSs signatures should also be carefully revised. For example, signature B (in Fig 1D) is the main component of mutagenesis in BRCA -/- cells and disappears in the absence of PolK. This signature is therefore a “Polk-dependent signature”. Signature C was only minimally contributing to mutagenesis in BRCA -/- cells but becomes the main contributor to mutagenesis once PolK is removed, therefore this signature is caused by an unknown polymerase or other process that becomes predominant in the absence of Polk.

In general, we did not name the signatures beyond identifiers such as 1A, 1B etc. The only exception is signature 1C which we also named Sig.POLK Δ , as it correlates with the deletion of pol kappa, and with lower polK expression in cancer. It should be emphasised that these signatures are primarily intended for the interpretation of the experimental data in this paper, and not for inclusion in the ever-expanding canon of mutational signatures in cancer.

Reviewer #2 (Remarks to the Author):

The manuscript by Chen et al is an interesting study of mutagenesis patterns in BRCA1 deficient cells that suggests a role for 53BP1 in antagonizing template switch recombination and promoting translesion synthesis (TLS). This topic is of broad scientific interest, particularly given the observation that BRCA-deficient cancers often exhibit a base substitution signature (SBS3), whose etiology has been difficult to pinpoint. The authors use engineered DT40 cell lines in conjunction with genomic sequencing analyses to evaluate the effect of gene disruption on mutation patterns in BRCA1 deficient cells. Whole genome sequencing analyses reveal a modest reduction in single base substitutions (SBS), but not indels, after deletion of TLS enzymes in BRCA1 deficient cells. The study goes on to show that increased spontaneous and cisplatin-induced mutagenesis in BRCA1 deficient cells is 53BP1 dependent. This 53BP1 associated effect is independent of Ku70-mediated NHEJ. Figure 4 shows a role for 53BP1 deficiency in restoring partial gene conversion activity induced by AID in a somatic hypermutation locus, but did not validate a role in promoting SBS in this locus. Thus, Figure 6 is vital to the conclusion, yet is underwhelming in showing a direct role in regulating TLS. Major critiques are listed below:

We thank the reviewer for emphasising the relevance of our results, and added a reference to the SBS3 signature to the abstract.

1) The authors should be transparent in their discussion regarding potential limitations of their sequencing method and analyses. Examination of the methods section suggests there was compromise of the sequencing data due to cross-species contamination that required more stringent filters and in some cases manual processing. This may reduce the sensitivity for structural rearrangements, such as tandem duplications and templated insertions that have been reported in BRCA1 deficient cells (e.g. PMID 32680986, 32680986). These technical limitations may have biased their analyses towards easier-to-detect SBS and small indel signatures.

The cross-contamination of samples is something we routinely experience, and it is probably a general problem in next generation sequencing. It happens at the sequencing facilities primarily within the flow cells, and we detect it more easily when we sequence chicken genomes, as the contamination is often with human DNA. If someone sends samples of human cells, contamination with further human DNA is in many instances undetectable.

We used two-step filtering to remove contaminating aligned reads: we filtered for the mapping quality of mutation-supporting reads, and we ignored mutation calls where there were jumps in the coverage on each side of the mutation, with precise criteria now added to the methods section. All filtering was algorithmic, there was no manual processing at all.

This filtering was only applied to detected SBS and short indel mutations (found by the IsoMut pipeline), so the filtering itself would not influence the detection of structural rearrangements. The current manuscript addresses the mechanism of formation of base substitutions and short indels, therefore we did not report large rearrangements at all (this is emphasised in the discussion). We are able to detect large rearrangements, and we reported these in BRCA mutant cells in our earlier papers (Zamborszky, 2017; Poti, 2019). BRCA1 deficient DT40 cells do not produce many structural rearrangements such as tandem duplications. However, we do have unpublished WGS data of cell lines which generate many TDs, which illustrates that our structural rearrangement detection pipeline (not used in the current manuscript) can detect these.

In addition to providing more detail on the filtering steps in the methods, we also clarified at the beginning of the results section that we only detected base substitutions and short indels.

2) Figure 5 interpretation requires 53BP1 reconstitution experiments to establish whether the observed differences in TLS proficiency are due to 53BP1 deficiency. This is particularly true since cytosolic fractions, that lack detectable 53BP1 protein, have distinct TLS activity.

We received help from the laboratory of Fabrizio d'Adda di Fagagna for such reconstitution efforts, who have joined the author list of the paper. Purification of the whole protein proved unsuccessful, probably because it is a very large protein with extensive disordered regions. However, we were able to test a C terminal fragment (residues 1053-1711) which contains most of the structured domains and is sufficient to form foci at double strand breaks in cells. We added this (which was available in limited quantities) to reactions which contained 25 ug cytosolic and 4 ug nuclear extract, as these showed the strongest difference based on the presence or absence of 53BP1. We tested two concentrations (10 nM and 50 nM). The 50 nM reactions showed partial reconstitution in terms of both replication efficiency and TLS%, which seems to make sense as we roughly estimated the concentration of 53BP1 in the nucleus around 20 nM (based on Beck et al., Mol Syst Biol 2011). We were unable to compare the reconstitution amounts by Western blot as none of the available antibodies recognised the 1053-1711 fragment.

These results, which have been incorporated into the manuscript and into Figure 6, suggest that the binding of 53BP1 is already sufficient to influence the choice of damage bypass pathways, without an obligate function for its the large disordered regions. However, due to the partial rescue effect, we do not wish to overemphasise the mechanistic conclusions of the reconstitution experiment.

3) The authors chose to focus their studies on BRCA1 rather than BRCA2. Did they examine the phenotype of BRCA2/53BP1 double knockout cells? This may help to clarify whether the role of 53BP1 in promoting TLS is directly related to or independent from its ability to restore HR-mediated repair.

We did not create any double mutants with BRCA2 primarily for technical reasons. All mutants included in this manuscript were generated by homologous gene targeting rather than e.g. CRISPR, removing large functional domains of the targeted genes. Homologous targeting does not work in BRCA1 or BRCA2 deficient cells, so all double mutants were made by disrupting the BRCA1 gene using homologous gene targeting in the respective single mutant lines, as described in the methods. However, we could not follow the same approach in the case of BRCA2, as its knockout constructs are particularly ineffective (removing most of this large gene). We could use CRISPR

to target 53BP1 in the BRCA2^{-/-} background, and we may try this in the future, but it would not produce a precisely equivalent 53BP1 mutant as used in this study.

Reviewer #3 (Remarks to the Author):

The study makes use of gene knockouts in chicken DT40 cells combined with whole genome sequencing to address potential contributing factors to the enhanced level of mutagenesis found in BRCA1 and BRCA2 deficient conditions, in particular in base substitutions. Some of the more interesting findings are i) Translesion synthesis contributes to part of the increased SBS rate, ii) Elevated levels of mutagenesis in HR compromised cells treated with cisplatin, arguing for error-free repair by HR in genetically non-compromised conditions, iii) the observation that 53BP1 apart from restricting gene-conversion in BRCA1 mutant cells (as expected) it also restricts the length of gene-conversion tracts (figure 4c).

I don't consider the study a very good candidate for publication in Nature communication. In my opinion, while I feel that the experiments are solid and well executed, the total is somewhat of a mixed bag, which added up does not provide clear (mechanistic) and conclusive insight into the underlying biology underlying the increased degree of SBS(3) in HR compromised cells. It is largely descriptive, which is not undervalued, but for this type of journal I would expect more substantial advance over what is already known. As an example: the authors show the involvement of TLS, but it would be rather unexpected if not given that TLS is involved in most SBS mutagenesis. The source of the SBS mutagenesis is still unknown, and why it is elevated in HR compromised cells as well.

We thank the referee for the comments on our manuscript. We would like to argue that the TLS-based origin of SBS3 is not trivial. There are numerous major sources of mutagenesis that are unrelated to TLS, such as normal DNA replication (its effects are best seen in mismatch repair deficiency, e.g. SBS6, SBS15, SBS20, SBS26), cytosine/methylcytosine deamination (SBS1, SBS2), miscoding by damaged bases (e.g. SBS18). In fact, TLS has only been implicated in a small minority of the mutagenic processes seen in cancer genomes, such as the effect of cisplatin (SBS31, SBS35) and even there the proof is partially lacking.

The notion that TLS takes over in the absence of HR is also not trivial, as TLS cannot resolve DNA breaks, prime targets for HR. The idea of increased TLS in HR mutants has been aired in the literature before, receiving partial proof in a recent yeast study which we are now citing (Loeillet, 2020 PNAS). We believe our data provide significant new evidence for the origins of mutagenesis in HR deficient cells.

REVIEWERS' COMMENTS

Reviewer #1 (Remarks to the Author):

The authors have addressed all my concerns in the revised version of the manuscript.

Reviewer #2 (Remarks to the Author):

The revised manuscript by Chen et al has been notably improved with the addition of some new experiments, as well as improvements in presentation clarity. The data are compelling to suggest a role for TLS in the elevated mutagenesis patterns observed in BRCA1/2 deficient cells. While the results may not be unanticipated given the predominant role for TLS in mutagenesis, this is the first formal demonstration of this functional relationship in the setting of BRCA deficiency. The additional putative role for 53BP1 in regulating the balance between TLS and template switching is intriguing and hypothesis generating. The authors have adequately addressed my concerns and I am supportive of acceptance for Nature Communications.